# Analysing Representations Through Layers: Token-Level Semantic Evolution in Clinical Language Model

## Abstract

Generative AI has significantly enhanced clinical decision-making and support for medical diagnosis. However, the black box nature of Large Language Models (LLMs), the lack of interpretability constrains their extensive use across clinical settings. This study develops and demonstrates a novel methodology that combines sparse autoencoders with token-level activation analysis to uncover and interpret layer-wise semantic evolution in clinical language model, enabling interpretable decision support in cancer text classification. The approach provides a representational interpretable technique to understand the underlying mechanisms of a domain-specific ClinicalBERT transformer to bridge the gap between the obscure nature of LLM model and human understandability. Sparse Autoencoders (SAEs) are employed to extract activation vectors and visualize the hidden embedding layers, offering deeper insights into how clinical concepts are encoded and transformed with the model. The experiments have been conducted using publicly available cancer text data as a case study on ClinicalBERT first four layers and last four layers, we observe steady advancements in feature adaptation on the last layers contained the task specific embeddings as compared to general feature adaptation in early layers. Lower layers capture syntactic and lexical patterns, while upper layers encode high-level clinical semantics. Whereas middle layers produced mixed, entangled representations making them unsuitable for stable token-level analysis. Therefore, we conducted a classification task using representation from first four and last four transformer layers to assess the interpretability of ClinicalBERT across its architecture. The model achieved 94% classification accuracy, indicating deeper layers capture highly discriminative features crucial for decision-level tasks. In contrast, the early layers yielded only 24% accuracy indicating limited representational capacity for such clinical classification. The key insights of these layers also demonstrate strong token level interpretability, reinforcing their empirical robustness in clinical applications.

## 1 Introduction

With the advent of Generative AI in the healthcare industry, AI tools are increasingly utilised by healthcare practitioners and pathologists to alleviate workload pressures and redefine workflows, thereby enhancing decision support (Andrew, 2024). Large Language Models (LLMs) are among the latest advancements in generative AI. LLMs represent a subset of deep learning models, pre-trained on extensive text data, designed to generate and comprehend text by identifying patterns within the data. Specifically, LLMs employ a neural network architecture known as the Transformers, which is engineered to sequentially process and generate textual data. Additionally, LLMs incorporate human feedback during training to ensure their responses closely align with user intent. Consequently, these models can engage in natural language conversations, perform language translations, generate various forms of written content, and answer questions across diverse subject areas (Brodnik et al., 2023; Edwards et al., 2025). Since LLMs are notoriously "black box" systems (de Carvalho Souza et al., 2025), meaning their inner mechanisms are obscure and difficult to decipher, ensuring the "interpretability of these models" remains a significant challenge. The lack of transparency in complex models makes it difficult to understand how they produce outputs and to

identify which features are important for decision-making tasks (Weidinger et al., 2022). Inadequate interpretability in language models can lead to the generation of misleading or erroneous content, often referred to as hallucinations. While the outputs could be syntactically plausible and contextually convincing, they may lack factual accuracy and semantic reliability (Liao & Vaughan, 2023). Encoder-only transformer-based language models have shown progress in clinical text classification (Chen, 2025). However, the mechanisms underlying these classifications and decision-making processes remain insufficiently underexplored. This study aims to track the feature activations in LLMs to elucidate the internal mechanism underpinning their classification task. Accordingly, we utilise Alsentzer et al. introduced ClinicalBERT, which has become foundational for many healthcare NLP applications (Alsentzer et al., 2019) and conduct layerwise analysis focusing on both upper and deeper layers to examine how feature activations contribute to interpretability within clinical NLP tasks. We propose a novel methodology to interpret representations for analysing layer-wise semantic evolution in transformer language models using sparse concept discovery. We introduce a novel framework for enhancing interpretability in transformer-based clinical language models, addressing the critical need for transparency in high-stakes domains such as disease diagnosis and policy-making. Unlike prior work that treats hidden representations as opaque, our method leverages Sparse Autoencoders (SAEs) to uncover concept neurons within token-level activations of ClinicalBERT. This approach yields a layer-wise, mechanistic view of semantic evolution, enabling the discovery of clinically grounded concepts directly from model internals. By aligning these sparse, interpretable features with domain-relevant constructs, our method transforms raw activations into cognitively accessible representations. This not only advances the interpretability of ClinicalBERT but also establishes a principled pathway toward trustworthy, concept-level explanations for clinical text classification tasks.

## 2 RELATED WORK

### 2.1 FEATURE EXTRACTION FROM DEEP LEARNING MODELS

The extraction of features aims to extricate the relevant features from the processed text data, leading to better efficiency and explainability. Language models like BERT have demonstrated impressive abilities in feature extraction methods like iAMP-AttenPred (Xing et al., 2023) for an antimicrobial peptide predictor to analyse peptide sequences. The extracted features from the final encoder layer are fed into the downstream models (CNN-BiLSTM-Attention). However, how BERT features influence and how they produce classification outputs are lacking in this framework. The embedding layers mechanism and attention weights should be analysed to identify functionally critical residues to track the interpretation of classification results. Another framework (Talebi et al., 2024) demonstrated that BERT variants deep contextual embeddings significantly outperform traditional models (eg. Random forest, XGBoost), especially in high imbalance protocol classes. Their explainability shows that the combination of word-level importance score with human evaluation with gradient-based attribution, to identify critical words. Unlike gradient-based post-hoc attributions, our model integrates explanation mechanisms inside the architecture by extracting embeddings-weighted concept features, which are tied to known clinically influenced tokens from specific hidden layers, enabling concepts mapping of model decisions to clinical knowledge.

### 2.2 INTERPRETABLE FRAMEWORKS

Interpretable machine learning for clinical text classification has gained significant attention in recent years due to its potential to enhance decision support in healthcare applications. Several studies have focused on developing models that not only achieve high predictive performance but also provide transparent and explainable outcomes for clinical practitioners. For instance, Ling (Ling et al., 2023) presented an interpretable machine learning framework for classifying clinical computed tomography (CT) reports, specifically focusing on temporal bone fractures. Their study emphasises the importance of integrating explainability into clinical natural language processing (NLP) pipelines to facilitate trust and adoption in medical settings. Chen (Chen et al., 2023) proposed an interpretable hybrid model for automated patient-wise categorisation of hypertensive and normotensive conditions using electrocardiogram (ECG) signals. building upon such approaches, this research proposed sparse representation technique on token-level feature activations to extract clinical relevant tokens to improve classification accuracy and reduce model opacity. These advancements

collectively aim to bridge the gap between complex neural architectures and the demand for transparent, human-understandable decision support tools in healthcare.

## 2.3 REPRESENTATION LEARNING

Representation learning in transformer-based language models has been widely explored through layer-wise analysis and probing methods. Subsequent probing studies have shown that information is concentrated in the upper layers of the BERT architecture introduced by (Devlin et al., 2019). In particular, research has demonstrated that averaging the last four hidden layers has become a common practice for capturing rich contextual embeddings (Tenney et al., 2019; Rogers et al., 2021; Liu et al., 2019), leading to their widespread use for downstream tasks such as classification and embedding extraction. Subsequent work has empirically validated this practice, showing that concatenating or averaging the last four layers yields stronger contextual embeddings than relying on a single layer alone (Hosseini et al., 2023). Building on this, Tenney et al. conducted a systematic probing study revealing that lower BERT layers primarily encode lexical and syntactic information. In contrast, higher layers specialise in semantics and task-specific reasoning (Tenney et al., 2019). This finding motivates our focus on comparing the first four and last four layers while excluding the middle layers, which tend to contain mixed representations that are less interpretable. Numerous studies have utilised hidden activations from ClinicalBERT for tasks such as clinical concept extraction, risk prediction and disease classification, but these have largely treated the model as a black box. However, these works have not addressed token-level semantic evolution in domain-specific models like ClinicalBERT. To bridge this gap, our study bridges the gap by integrating SAEs with token-level activation analysis to examine how semantic information evolves across the ClinicalBERT layers. Unlike prior work, we provide a representational interpretability of token representations within cancer-related text, hence advancing interpretable decision support for medical classification tasks.

## 3 PROPOSED METHODOLOGY

Our proposed methodology introduces a token-level mechanistic interpretability framework that combines Sparse Autoencoders (SAEs) with ClinicalBERT hidden layer activations to uncover layer-wise semantic evolution. By analyzing activations from the first and last four layers as shown in Figure 1, the approach enables transparent cancer text classification and provides clinician-friendly visualization and decision support.

### 3.1 TEXT DATASET AND TEXT PRE-PROCESSING

We employ the Cancer Abstract Dataset (Hossain & Nuzhat, 2024), which comprises medical text tailored to cancer-related documents alongside a set of generic clauses. The dataset is originally annotated into four categories: Thyroid Cancer, Colon Cancer, Lung Cancer, and Generic. Since the objective of this study is to advance interpretability in clinical text analysis, we omit the Generic category during preprocessing. This refinement ensures that the dataset is aligned with clinically meaningful labels, thereby making it more suitable for downstream tasks in clinical text classification and for integration into clinical NLP pipelines. Text pre-processing is a critical step in preparing unstructured textual data for feature extraction and downstream tasks. This text contained raw text that included various forms of noise, such as irrelevant punctuation and non-informative tokens. To address this, the pre-processing steps involved stemming that reduces words to their root forms by stripping affixes, although this process may result in non-dictionary tokens, lemmatization leverages the linguistic context to convert words into their canonical dictionary form while preserving the semantic integrity, and applied stop words removal to focus on meaningful elements from clinical text, thereby enhancing downstream tasks such as concept extraction and clinical text classification. It enhances the tokenisation process, ensuring that the textual data is optimally structured for ClinicalBERT. These processes aim to normalise the text, reduce dimensionality, and improve the generalisation capacity of learning models. This normalisation step is crucial for models such as transformers and other deep learning architectures require numeric vectorisation for training and inference.

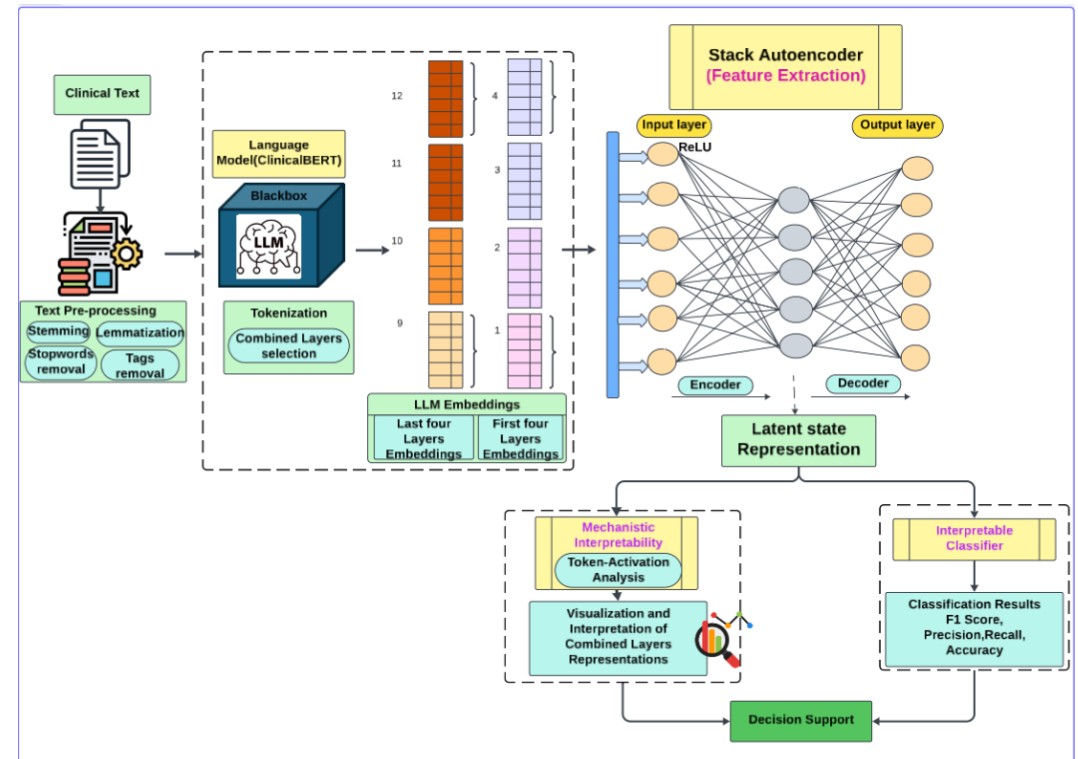

Figure 1: Token Level Semantic Evolution in Clinical Language Model

## 3.2 LAYER-WISE REPRESENTATION EXTRACTION

The evolution was done on an encoder-based transformer model, ClinicalBERT 12-layer architecture where each layer contained hidden information. The layers are chosen for interpretability of the language model as they represent hidden activations inside each layer and contain hidden information related to the behaviour of the concepts. This rationale for the study is defined in research (López-Otal et al., 2025), which illustrates that BERT layers may contain more complex linguistic patterns captured by deeper and deeper layers and may exhibit pipeline behaviour. The experiment is initiated to examine the representational evolution of clinicalBERT by analysing its upper four layers and deeper four layers. This investigation aimed to illustrate how these layers contribute in clinical decisions, with a focus on the semantic and discriminative behaviour of each layer group. Tokenisation is applied to the input text to facilitate the extraction of hidden state parameters from successive hidden layers of the model. After tokenisation of text data, the token matrix was passed through all 12 layers. The activation matrix, which is called the output of each layer using the hidden-states variable from all 12 layers, the hidden states of the sum of the first four layers and the last four layers were extracted for token activations. The sum of the upper layers and deeper layers is averaged, and a single representative activation vector is produced. This averaged vector is subsequently used as input to the autoencoder deep learning model, which extracts bottleneck features corresponding to specified hidden layer configurations. The equation of combined layer combinations of hidden BERT token embeddings is shown as:

$$S = \sum_{l=n}^{12} H^{(l)} \tag{1}$$

In equation 1, $n$ denotes the number of hidden layers from 12 layers, and $H$ denotes the hidden states, which are 768 in ClinicalBERT, and $l$ refers to the number of upper and deeper layers, we assumed 4 for upper four layers and and -4 for deeper layers.

## 3.3 SPARSE CONCEPT REPRESENTATION LEARNING

The Sparse Autoencoder (SAE) is adapted to learn concept activations that make ClinicalBERT more interpretable. The idea is to break down the dense and complex BERT embeddings into a sparse set of meaningful features, where each feature represents a distinct clinical concept. The standard encoder and decoder equations for sparse autoencoder (SAE) are adapted to the task of learning "concept activations" for interpreting ClinicalBERT. The aim is to decompose dense, uninterpretable activation vectors into a sparse combination of meaningful features where each feature represents a distinct concept. The SAE architecture includes an encoder with an input size of 768 (the size of BERT's token embeddings) and a hidden layer of 1024 neurons, which acts as the concept layer. This over-complete layer allows capturing richer patterns while enforcing sparsity, so only a few neurons activate for each input. The decoder reconstructs the original 768-dimensional embedding from these sparse concept activations. To prepare input, we first take a clinical text and compute token embeddings by summing the last (or first) four hidden layers of ClinicalBERT, resulting in a 768-dimensional vector for each token. These embeddings capture deep contextual information from all 12 layers of the model. The encoder then maps this vector $x \epsilon R^{768}$ to a latent representation $z$ using the equation:

$$z = ReLU(W_e.x + b_e) \tag{2}$$

In equation 2 W$e$ is the encoder weight matrix, b$e$ is the encoding bias vector and the rectified linear unit(ReLU) is a non-linear activation function which helps achieve sparsity. The decoder function is to take the sparse latent code z and reconstruct an approximation of the original input $\hat{x}$. The decoder reconstructs the original embedding using:

$$\hat{x} = (W_d.z + b_d) \tag{3}$$

In equation 3 $\hat{x} \epsilon R^{768}$ aims to approximate the original input $x$. To model is trained to minimise the loss function that balances the reconstruction accuracy and sparsity of activations. The total loss is defined as:

$$L = MSE(x, \ \hat{x}) + \lambda ||z||_1 \tag{4}$$

Here, MSE (Mean Squared Error) ensures faithful reconstruction, while the L1 term promotes sparse activations. The SAE is trained on Adam optimiser with the learning rate of $2e^{-5}$ over 20 epochs, typically achieving a loss around $10^{-3}$, illustrated in equation 4 and $\lambda$ is the learning rate and $\lambda=10^{-3}$. Once trained, the latent neurons (concept neurons) in $z$ can be analysed to identify meaningful clinical concepts. By observing which neurons activate strongly for each cancer class, we can visualise and interpret them as high-level patterns thus turning ClinicalBERT's hidden states into human-understandable features.

## 3.4 TOKEN ACTIVATION ANALYSIS

After the training of SAEs, the next step is to run experiments and make it interpretable in terms of human-understandable. The token-level activations were performed for selected text of the clinical text class of the dataset, for instance, text from Thyroid cancer, Colon cancer, and Lung cancer types. To allow the interpretable representations of each learned feature for deeper analysis. The model distinguishes three cancer classes, each associated with distinct activation patterns that correspond to class-specific semantic concepts. These visualised activations provide interpretable cues, enabling domain experts to explore the underlying representations of each class. Such conceptual mappings support decision-making by offering clearer insights into the classification rationale, as demonstrated in the results. The steps involved for the analysis of each cancer type are:

1. Tokenisation of selected class-wise sentences and passing through a specific combination of layers mechanism.

2. Extracting hidden state activations from specified layers to understand how these layers work during decision making.

3. Passing these activations through trained SAE to get concept bottleneck activations for selected features index.

4. Tracking these activation values in the format of tokens for each cancer class concept.

5. Classification report of each cancer type by following the activation values of the sum of the upper layers and lower (4) layers.

The token activations are visualised using a plotting graph to allow meticulous analysis of which token activated on given features. By testing various class-specific sentence types, we gained insights into the linguistic and semantic concepts represented by each feature index, consisting of both general-purpose expressions and those rich in domain-specific terminologies.

# 4 RESULTS

## 4.1 EXPERIMENTAL SETUP

### 4.1.1 FEATURES EXTRACTION FROM SAE

The trained sparse autoencoders provided a detailed interpretive lens into the latent concepts encoded within the ClinicalBERT model by analysing activations across both the sum of the first four embedding layers and the sum of the last four layers. The analysis revealed a clear evolution of feature representations, supporting the hypothesis that the early layers capture more general linguistic and syntactic features, while the later layers encode task-specific and domain-specialised information. Furthermore, the comparative assessment between summed first/last four layers of domain-adapted ClinicalBERT demonstrated distinct shifts in internal representations, highlighting how fine-tuning on clinical data alters layer-wise activations to better align with the requirements of clinical NLP tasks.

### 4.1.2 LAST FOUR EMBEDDING LAYERS INTERPRETABILITY

The last four layers of ClinicalBERT LLM predominantly activated on semantically related and domain-specific terms. After training the model, selected test sentences are passed through it to identify and extract the specific tokens that activate interpretable concept neurons. Some tokens like [CLS], called classification tokens in BERT-based models, play a critical role, especially when working with later layers. It's a special token added to the beginning of every input sequence. This hidden state of [CLS] is designed to capture a summary representation of the entire input sequence. We observed that [CLS] is active in later layers, which illustrates that the embedding layer is ready for a classification task. The deeper layers are making semantic relationships and task-specific concepts (for instance, disease, symptoms, reasoning relationships) from a clinical perspective as we are evolving the specialised ClinicalBERT. The "genetic", "epigenetic", and "disease" are referring tokens in colon cancer as shown in Figure 2. The tokens "lung", "cancer", "deaths", and "increasingly" are activated in lung cancer illustrated in Figure 3. Although other tokens like "morphofunctional", "thyroid", "immunohistochemical" are semantically related to each other to take part in the decision in Thyroid cancer text interpretability. Similarly, the "lung", "cancer", "malignant" and "neoplasm" are activated and semantically related for lung cancer shown in Figure 4 in the clinical domain.

### 4.1.3 FIRST FOUR EMBEDDING LAYERS INTERPRETABILITY

The first four layers of the language model build progressively abstract representations of the input text. These layers usually capture immediate context, parts of speech and begin to capture short clinical events and logic patterns, as shown in the bottom part of each cancer type. The [SEP] token in BERT-based language models is represented as a special token used in NLP tasks and machine learning (ML) models to mark the separation among different text segments. It is somehow activated in the first four layers in colon and lung cancer. Its activation in early layers reflects the importance of knowing the common word as [SEP], where segments start and end. this is less relevant in later layers. We observed that less-important words like "suggest", "single", "least", "three" and activated in colon cancer. The "commonest", "world", "accounts" are activated more and there is no clue of important tokens like "lung" and "cancer" depicted in the bottom part of Figures 2, 3, and 4.

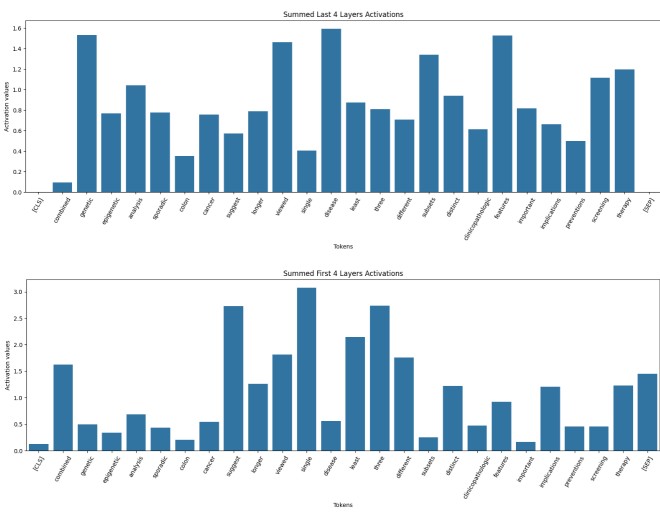

Figure 2: Token Activations of last four and first four layers on Colon Cancer

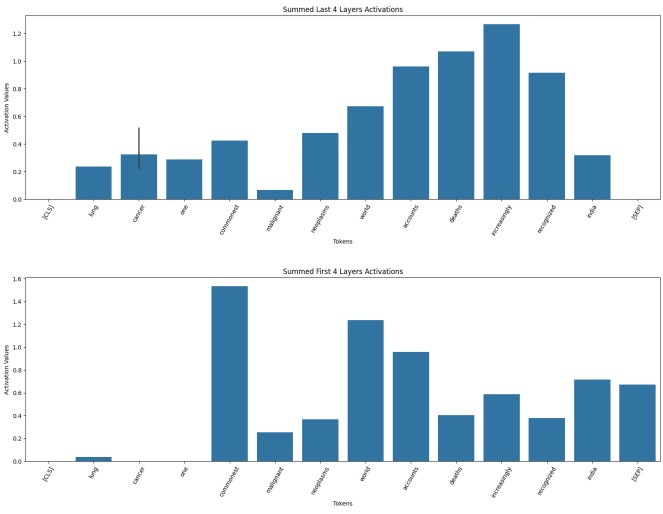

Figure 3: Token Activations of last four and first four layers on Lung Cancer

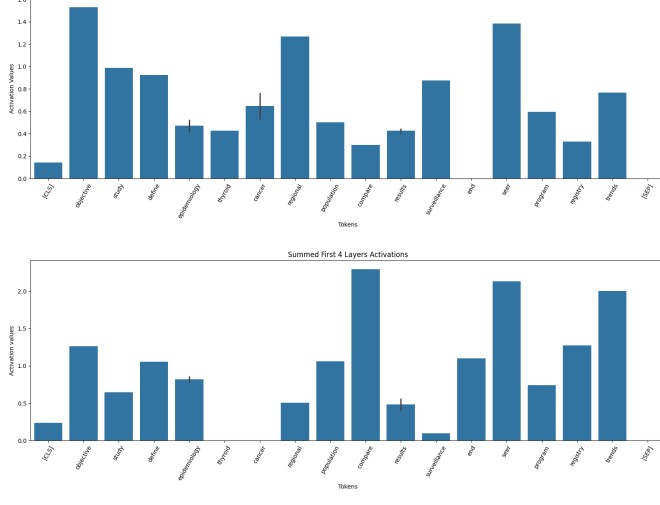

Figure 4: Token Activations of last four and first four layers on Thyroid Cancer

## 4.2 CLASSIFICATION RESULTS TO EVALUATE INTERPRETABILITY

To assess the interpretability of token activations within ClinicalBERT, a comparative analysis was conducted using embeddings derived from the first/last four transformer layers. Classification is employed to evaluate concept activations by determining whether the activated tokens are supporting for model decisions. Logistic regression was employed to classify multiple cancer types, with an 80/20 train-test split. The results, summarised in Table 1, indicate that activations from the final four layers significantly contribute to disease classification performance. Specifically, the model achieved precision and recall scores of 96% and 92% for thyroid cancer, 93% and 90% for colon cancer, and 90% and an exceptional 99% for lung cancer, respectively. These findings suggest that deeper-layer representations are more informative for clinical decision-making. In contrast, Table 2 presents the classification performance based on activations from the initial four transformer layers. The results indicate that these upper-layer representations offer limited support for clinical decision-making. Specifically, the precision and F1-score for colon cancer were 22% and 26%, respectively, while lung cancer achieved 28% precision and 35% F1-score. Thyroid cancer exhibited the weakest performance, with precision and F1-score values of 19% and 16%, respectively. These findings underscore the diminished diagnostic utility of early-layer embeddings in ClinicalBERT. Further analysis, presented in Table 3, reveals that embeddings from the last four layers yielded an overall classification accuracy of 94%, whereas those from the initial four layers resulted in a substantially lower accuracy of 24%. This disparity underscores the heightened relevance of deeper transformer layers in capturing clinically salient features and enhancing model interpretability in multi-class disease classification tasks.

Table 1: Last Four Layers Classification Result

| Cancer Types | Precision | Recall | F1-score |
|---|---|---|---|
| Colon Cancer | 93% | 90% | 92% |
| Lung Cancer | 90% | 99% | 94% |
| Thyroid Cancer | 96% | 92% | 94% |

Table 2: First Four Layers Classification Result

| Cancer Types | Precision | Recall | F1-score |
|---|---|---|---|
| Colon Cancer | 22% | 32% | 26% |
| Lung Cancer | 28% | 47% | 35% |
| Thyroid Cancer | 19% | 15% | 16% |

Table 3: Classification result accuracy on Last four layers and First four layers

| Selected Layers mechanism | Accuracy Cancer Types |
|---|---|
| Last Four Layers | 94% |
| First four Layers | 24% |

## 5 DISCUSSION

Our findings reveal that the last four layers of ClinicalBERT capture richer and more clinically meaningful token-level semantics compared to the first four layers, consistent with prior transformer analyses that early layers encode shallow syntactic features while deeper layers specialise in task-relevant semantics. Notably, our token-level activation analysis demonstrated that semantic evolution occurs progressively across layers, with early layers showing diffuse activations and later layers exhibiting sharper, domain-specific patterns related to cancer concepts. This layer-wise interpretability extends beyond post-hoc explanations like SHAP or attention visualization, offering a direct window into the inner workings of domain-specific language models. These insights have

implications for building safer and more transparent clinical AI systems, as they allow clinicians to trace model decisions back to interpretable latent concepts.

## 6 CONCLUSION AND FUTURE WORK

This study investigates the internal mechanisms of transformer-based language models, specifically focusing on feature extraction of each class and how token-level representations contribute to decision support in clinical contexts. An analysis of domain-specific ClinicalBERT reveals that domain-specific information becomes increasingly salient in the upper layers of the architecture, with later layers encoding more clinically relevant features compared to earlier ones. Furthermore, aggregating representations across layers yields a unique embedding profile, highlighting how different layers collectively capture and preserve semantic information. Classification is performed using the concept activations for each cancer class to assess whether the activated tokens predominantly contribute to the text classification task. Subsequent experiments extend the analysis to a broader spectrum of large language models (LLMs), including GPT models, T5, and domain-specific adaptations of BERT. Deep learning models would be applied for grasping the final classification reports for indepth explainability of language models. These investigations aim to elucidate the internal dynamics of hidden embedding layers and their roles in decision-making processes. Furthermore, knowledge graphs are proposed as a means of integrating expert-level domain knowledge in a structured format, thereby enhancing semantic precision and mitigating the hallucination tendencies observed in LLM-generated outputs.

### AUTHOR CONTRIBUTIONS

If you'd like to, you may include a section for author contributions as is done in many journals. This is optional and at the discretion of the authors.

### ACKNOWLEDGMENTS

Use unnumbered third level headings for the acknowledgments. All acknowledgments, including those to funding agencies, go at the end of the paper.

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

## A APPENDIX

We used ChatGPT, a large language model developed by OpenAI, to aid in improving the clarity and readability of the manuscript text in some subsections. The model was not involved in generating research results or making scientific decisions.

