# OpenReview forum: "Analysing Representations Through Layers: Token-Level Semantic Evolution in Clinical Language Model"
_ICLR.cc/2026/Conference — Submitted to ICLR 2026_

### Official Review · Reviewer_ztKu · 2025-10-26

**Soundness:** 2
**Presentation:** 3
**Contribution:** 2
**Rating:** 2
**Confidence:** 5

**Summary:**

This paper introduces a methodology to interpret layer-wise token representations in ClinicalBERT by combining Sparse Autoencoders (SAEs) with token-level activation analysis. The authors aim to uncover how semantic features evolve across layers in a domain-specific transformer used for clinical text classification, specifically on cancer-related documents. Using the Cancer Abstract Dataset, they analyze the first and last four layers of ClinicalBERT, applying SAEs to extract sparse, interpretable concepts that map activation patterns to clinical tokens. The study finds that deeper layers encode domain-specific semantics and yield 94% classification accuracy on cancer subtype identification, compared to 24% for early layers. Visualizations of token activations show that upper layers capture clinically meaningful terms (e.g., lung, cancer, malignant), while lower layers capture lexical or syntactic noise. The authors argue this provides a mechanistic interpretability framework to bridge “black box” LLMs and human reasoning in clinical decision support

**Strengths:**

- The paper tackles a timely question in clinical NLP with a focus on transparency and trust in high-stakes decision-making.
- Integrating Sparse Autoencoders with layer-wise activation analysis offers a well-motivated, concept-level interpretability approach distinct from post-hoc attribution methods (e.g., attention maps, SHAP).
- The explicit comparison of first vs. last four layers aligns with existing BERTology findings and gives empirical grounding to the claim that semantics evolve hierarchically.

**Weaknesses:**

- The experimental setup isolates only the first and last four layers, omitting the middle-layer transition dynamics where semantic disentanglement likely occurs. A full 12-layer evolution trajectory would provide stronger evidence of “semantic evolution.”
- There are no benchmarks against alternative interpretability approaches (e.g., probing classifiers, integrated gradients, concept bottleneck models), leaving unclear whether SAEs offer superior conceptual disentanglement or consistency with ohter methods.
- The Cancer Abstract Dataset is relatively narrow and synthetically clean. There is no evaluation on real-world, noisy EHR narratives or external clinical datasets to test the method’s generalizability.
- The interpretability claims are primarily qualitative and descriptive. Metrics like concept purity, mutual information between discovered neurons and clinical categories, or layer-wise alignment scores would substantiate claims of “interpretable concept discovery.”
- The study trains SAEs directly on representations derived from ClinicalBERT and then uses those same embeddings for downstream classification. This coupling raises questions about whether the SAE contributes interpretability or simply reprojects separable features.
- The methodology shows correlation (later layers produce better accuracy and semantic clarity) but does not establish causality between layer depth and concept disentanglement.
- The 94% accuracy and 24% early-layer baseline are reported without confidence intervals or multiple random seeds. Given the small dataset, overfitting or optimistic bias cannot be ruled out.
- The “interpretable” concepts are not validated by clinicians or domain experts, so interpretability remains algorithmic rather than human-grounded.
- Authors can benefit from trying other encoders (https://arxiv.org/abs/2504.03964, https://arxiv.org/abs/2506.10896) to see whether different encoders that see the same data are providing similar important features. They might provide similar observations to that in (https://arxiv.org/abs/2411.01322)

**Questions:**

- How robust are the results across different seeds, SAE architectures, or sparsity regularization strengths (λ in Eq. 4)? Does interpretability degrade under less constrained sparsity?
- Why were the middle four layers of ClinicalBERT omitted from analysis? Given their mixed representations, quantitative metrics could still reveal transitional dynamics crucial for semantic evolution.
- How does the proposed SAE-based concept decomposition compare to simpler dimensionality reduction techniques (e.g., PCA, ICA) or to established probing approaches in terms of interpretability and classification performance?
- Can the discovered “concept neurons” be directly mapped to known medical ontologies (e.g., SNOMED-CT, UMLS)? If not, what makes them clinically interpretable beyond token co-activation patterns?
- How does the method scale to longer clinical documents or to full patient-level records? ClinicalBERT is constrained to a squence length of 512 tokens. The authors can explore Clinical ModernBERT, Clinical BigBird, Clinical Longformer if given the time. Are there memory or computational constraints that limit applicability?
- Have the authors tested generalization to other datasets (e.g., MIMIC-III discharge summaries) to confirm that layer-wise interpretability patterns hold across clinical corpora?
- Can the authors provide quantitative measures of interpretability (e.g., sparsity ratio, token-activation entropy, neuron-level concept coherence) to complement qualitative token plots?

---

### Official Review · Reviewer_f1ZE · 2025-10-27

**Soundness:** 1
**Presentation:** 2
**Contribution:** 1
**Rating:** 0
**Confidence:** 4

**Summary:**

This paper proposes a method to interpret the internal representations of ClinicalBERT by applying Sparse Autoencoders (SAEs) to the hidden state activations of its early and late layers. The goal is to decompose dense embeddings into sparse, interpretable "concept" features for a three-class cancer text classification task. The authors claim that this method offers a direct window into the model's token-level semantics, demonstrating a progression from syntactic to semantic features across layers.

**Strengths:**

The paper addresses the important problem of model interpretability, particularly within the high-stakes domain of clinical AI.

The overall methodology is straightforward and generally easy to follow.

**Weaknesses:**

Lack of Novelty: The core methodology, using Sparse Autoencoders to find interpretable features in a model's activations, is a well-established technique in the field of mechanistic interpretability. The paper presents this as a novel framework but fails to cite, acknowledge, or build upon the extensive and highly relevant prior work in this area (e.g., the body of work from Anthropic and others).

Trivial and Unsurprising Findings: The paper's primary conclusions are not new knowledge. The findings that later layers in a transformer are more task-specific and semantically rich than early layers, and that these later layers yield higher classification accuracy, are foundational and well-documented properties of BERT-like models. Furthermore, the "interpretable insights" presented are trivial, such as discovering that the token "lung" is important for classifying texts about lung cancer. This does not demonstrate the "deeper insights" the paper claims to provide.

Critically Insufficient Literature Review: The related work section is extremely sparse and fails to position the work adequately. It omits entire subfields relevant to the paper's claims and cites only a handful of specific application papers instead of seminal or survey works. This reflects a lack of engagement with the current state of the art.

Weak Experimental Design and Evaluation:

Single Dataset: The analysis is confined to a single dataset, limiting the generalizability of the claims. Standard practices like cross-validation are not mentioned.

Conflating Accuracy with Interpretability: The paper incorrectly frames classification accuracy as a primary metric for "evaluating interpretability." High accuracy in later layers is an expected outcome of model training, not a measure of how interpretable those layers are.

Superficial Analysis: The token-level analysis is superficial, limited to showing bar plots for single examples without any systematic or quantitative evaluation of the discovered features.

Low Clinical Relevance: The chosen task, classifying abstracts into one of three obvious cancer types, is a toy problem with little to no real-world clinical utility. It is unclear what a domain expert would learn from interpreting a model for such a simple task, which undermines the paper's central motivation of providing decision support in clinical settings.

Poor Presentation and Technical Errors: The manuscript is unpolished. The writing is repetitive and contains grammatical errors. Figures are low-resolution with illegible axes, and one figure contains an unexplained plotting artifact (a single error bar).

**Questions:**

The paper's central findings (e.g., later layers are more semantic, the word "lung" is key for lung cancer) are well-established or predictable. Can you highlight a single, non-obvious clinical insight gained from your method that would not be apparent from a simple keyword analysis?

Given the simplicity of the classification task, could you elaborate on a realistic clinical scenario where the insights from your framework would provide actionable and useful information to a domain expert?

---

### Official Review · Reviewer_xnfK · 2025-10-30

**Soundness:** 3
**Presentation:** 1
**Contribution:** 1
**Rating:** 2
**Confidence:** 4

**Summary:**

This paper applies Sparse Autoencoders (SAEs) to analyze token-level activations in ClinicalBERT across different layers, focusing on cancer text classification. The authors extract representations from the first and last four layers of the 12-layer architecture, train SAEs to learn sparse concept representations, and evaluate interpretability through classification tasks. Their results argue that that deeper layers capture more clinically meaningful semantics.

**Strengths:**

1. Clinical NLP interpretability is important for high-stakes medical decision-making, making the motivation sound.

2. The layer-wise comparison provides a structured framework for analysis, with reproducible steps outlined. And the quantitative evidence supports the hypothesis about layer-wise semantic evolution.

**Weaknesses:**

1. The core result, that early layers encode syntactic features while deeper layers capture semantic and task-specific information, is well-established in transformer interpretability literature. The authors acknowledge this prior work but do not clearly articulate what new insights their analysis provides beyond confirming known behaviors in the clinical domain.

2. The approach combines existing techniques (SAEs + layer-wise probing). Standard SAE architecture with L1 regularization and MSE reconstruction loss offers no advances over prior interpretability methods. What specifically does this framework enable that existing approaches (attention visualization, gradient-based attribution, or standard probing) cannot? And why not analyze all 12 layers to show the complete semantic evolution trajectory?

3. The token activation plots are difficult to parse and don't provide clear mechanistic insights. How do clinicians use these visualizations for decision support?

4. Figures 2-4 are very hard to read and should be visualized better.

**Questions:**

See above.

---

### Official Review · Reviewer_CRkM · 2025-11-02

**Soundness:** 1
**Presentation:** 1
**Contribution:** 1
**Rating:** 2
**Confidence:** 3

**Summary:**

This paper is focused on understanding the representations of clinical language models. It investigates the interpretability of ClinicalBERT by analyzing token level semantic evolution across transformer layers using Sparse Autoencoders. The paper compares representations from the first 4 layers versus the last 4 layers of ClinicalBERT on a cancer text classification task. The methodology combines SAEs with layer-wise activation analysis to extract interpretable concept features. Experimental results on a cancer dataset show that the last four layers achieve 94% classification accuracy compared to 24% for the first four layers, supporting the hypothesis that deeper layers encode more clinically relevant semantic information.

**Strengths:**

- The combination of SAEs with token level activation analysis for clinical domain language models is interesting and provides a mechanistic view of semantic evolution across layers.
- The semantic evolution is demonstrated empirically with the difference between first and last layers performance on the cancer text classification task.
- The paper builds on recent interpretability research in LLMs and addresses interpretability in clinical NLP which is vital for healthcare applications.

**Weaknesses:**

- The finding that early BERT like model layers capture syntactic features while later layers capture semantic task features is well established (Rogers et al., 2021). This paper primarily validates known phenomena in a clinical domain rather than introducing fundamentally new.
- Similarly, SAEs have been widely used recently for mechanistic interpretability of large language models.
- The paper writing can be significantly improved. For instances, (i) lines 24-28 in abstract seem to contain two sentences in one. (ii) mathematical notations such as in line 231 does not have the variable x in math environment, (iii) different tokens in lines around 305 are not in consistent quotes, (iv) Figures 2-4 have bigger bars, but smaller text, (v) Eqn 1 and its description in lines 214-15 have confusing notations on layers.
- Overall the paper's methodological contributions appear applicative to healthcare applications, and the writing can be improved significantly.

References:
Rogers, Anna, Olga Kovaleva, and Anna Rumshisky. "A primer in BERTology: What we know about how BERT works." Transactions of the association for computational linguistics 8 (2021): 842-866.

**Questions:**

na

---

### Meta-Review · Area_Chair_z7aN · 2025-12-05

**Summary:**

All reviewers raised major concerns with this work, mainly in terms of findings not being novel or specific to clinical applications and no technical novelty.

**Reviewer Concerns:**

The authors provided no response.

**Reviewer Scores:**

I don't believe any rebuttal would have changed the outcome for this work and I encourage the authors to carefully review the suggestions from the reviewers and look at recent work combining SAEs and LLMs for interpretability.

---

### Decision · Program_Chairs · 2026-01-26

Reject